# Assessment of Quality of Life and Psychological Well-Being in Italian Adult Subjects with Prader–Willi Syndrome Using the Health Survey Short Form and the Psychological General Well-Being Index Questionnaires

**DOI:** 10.3390/healthcare13020158

**Published:** 2025-01-15

**Authors:** Anna Guerrini Usubini, Michela Bottacchi, Adele Bondesan, Diana Caroli, Graziano Grugni, Gianluca Castelnuovo, Alessandro Sartorio

**Affiliations:** 1Experimental Laboratory for Auxo-Endocrinological Research, Istituto Auxologico Italiano, Istituto di Ricovero e Cura a Carattere Scientifico (IRCCS), 28824 Piancavallo-Verbania, Italy; a.bondesan@auxologico.it (A.B.); d.caroli@auxologico.it (D.C.); g.grugni@auxologico.it (G.G.); sartorio@auxologico.it (A.S.); 2Psychology Research Laboratory, Istituto Auxologico Italiano, Istituto di Ricovero e Cura a Carattere Scientifico (IRCCS), 28824 Piancavallo-Verbania, Italy; m.bottacchi@auxologico.it (M.B.); gianluca.castelnuovo@auxologico.it (G.C.); 3Department of Psychology, Catholic University of Milan, 20123 Milan, Italy

**Keywords:** Prader–Willi syndrome, quality of life, health-related quality of life, psychological well-being, Health Survey Short Form-36, Psychological General Well-Being Index, rare diseases

## Abstract

**Background/Objectives**: Prader–Willi syndrome (PWS) is a rare, genetically determined neurodevelopmental disorder. Individuals with PWS face numerous challenges that significantly impact their psychological well-being and quality of life, ultimately limiting their personal and social functioning. This study aimed to evaluate the quality of life and psychological well-being in a sample of Italian adult patients with PWS compared to an age-matched control group of normal-weight Italian individuals. **Methods**: Thirty patients with PWS (11 men and 19 women; mean age ± SD: 36.4 ± 10.31 years; mean Body Mass Index (BMI: 35.7 ± 8.92 kg/m^2^) and thirty Italian adult individuals from the general population (5 men and 25 women; mean age ± SD: 32.1 ± 6.86 years; mean Body Mass Index (BMI: 21.8 ± 2.90 kg/m^2^) were studied. Quality of life and well-being were assessed using the Italian versions of the 36-item Health Survey Short Form and the Psychological General Well-Being Index. **Results**: Normal-weight subjects scored significantly higher than PWS patients on the physical health (*p* < 0.001) and social functioning (*p* = 0.047) subscales of the SF-36. Conversely, PWS patients scored higher on the vitality subscale (*p* < 0.001). Similarly, the vitality subscale of the PGWBI was significantly higher in PWS patients than in controls (*p* = 0.010), whereas the Self-Control subscale of the PGWBI was higher in controls compared to PWS patients, albeit not statistically significant (*p* = 0.057). **Discussion**: Patients with PWS exhibited impairments in various aspects of quality of life and psychological well-being, including physical, behavioral, and social domains. However, the higher vitality scores observed in PWS patients suggest a preserved dimension of their psychological well-being. **Conclusions**: These findings enhance the understanding of the psychological condition of patients with PWS and provide valuable insights for improving multidisciplinary psychological treatment approaches for these individuals.

## 1. Introduction

Prader–Willi syndrome (PWS) is a rare, genetically determined neurodevelopmental disorder caused by the failure to express paternally derived genes in the PWS critical region on chromosome 15 (15q11.2-q13). Specifically, PWS can result from a deletion in this region, maternal uniparental disomy of chromosome 15, defects in the imprinting center, or other abnormalities involving chromosome 15 [1]. The condition affects males and females equally across all ethnic groups, with an estimated prevalence of 1:10,000–1:30,000 cases in the general population, representing the most common genetic cause of syndromic obesity [2].

PWS is a complex disorder characterized by hypotonia and failure to thrive during the neonatal period, followed by hyperphagia leading to morbid obesity from early childhood (if not managed). Additional features include multiple endocrine disturbances, such as growth hormone [GH]/insulin-like growth factor-1 axis dysfunction, hypogonadism, hypothyroidism, and central adrenal insufficiency, as well as developmental difficulties like learning disabilities and typical dysmorphic features [3]. Psychiatric impairments are also common and include behavioral disturbances, anxiety, obsessive-compulsive disorder, depression, aggressive behaviors, skin picking, ADHD, psychotic symptoms, and cognitive impairment [4,5]. Such a phenotype seems to be related to hypothalamic dysregulation [6].

The numerous challenges faced by individuals with PWS—particularly the need for strict control of food intake and the presence of cognitive and behavioral disorders—significantly impact their psychological well-being and quality of life, often limiting their personal and social functioning. In a study conducted by Caliandro and colleagues [7], the quality of life in a sample of patients with PWS was compared to that of healthy subjects. Results indicated significant impairments in various physical, social, and emotional domains of quality of life in patients with PWS compared to their healthy counterparts. This study further divided the patients into two subgroups: those older than 14 years and those 14 years or younger, and compared each group with age-matched healthy controls. For patients older than 14 years, all subscales of the Short Form-36 used to assess quality of life showed significantly worse scores in PWS patients than controls, except for the vitality subscale, which did not differ significantly. For patients 14 years or younger, most subscales of the Child Health Questionnaire-Parent Form-50 (CHQ-PF50) also revealed significantly worse scores in PWS patients than controls, except for the bodily pain and family cohesion subscales, which were not significantly different.

While numerous studies [8,9,10] have shown that the quality of life in individuals with PWS is substantially impaired due to deficits across various life domains, there is limited evidence comparing quality-of-life data between PWS patients and healthy counterparts in the Italian context. Such comparisons can provide valuable insights into the clinical aspects of PWS and inform more effective treatment strategies. This study specifically aimed to evaluate the quality of life and psychological well-being in a sample of Italian adult PWS patients compared to an age-matched sample of normal-weight individuals.

## 2. Materials and Methods

### 2.1. Participants

The sample was composed of thirty patients with PWS (11 men and 19 women; mean age ± SD: 36.4 ± 10.31 years; mean Body Mass Index (BMI) ± SD: 35.7 ± 8.92 kg/m^2^). All patients with PWS showed the typical clinical phenotype [3]. Twenty-four subjects had interstitial deletion of the proximal long arm of chromosome 15 (del15q11-q13), while six patients had uniparental maternal disomy for chromosome 15. Seven subjects had obstructive Sleep Apnea Syndrome (OSAS); five of them were in treatment with Continuous Positive Airway Pressure (C-PAP); twelve patients had diabetes, five had hypertension, ten had dyslipidemia, and one subject suffered from epilepsy. Eleven subjects with PWS lived only with their parents, five with their parents and one or more brothers/sisters. One subject with PWS lived with their parents and the grandmother, seven with only one of the parents, and three with one of the parents and one or more brothers/sisters. In the remaining cases, one subject with PWS lived with the uncles, one lived alone, and one lived in a residential structure.

The mean MMSE score of patients with PWS was 26.6 ± 1.92, with all subjects having a score >24. No patients were under psychological treatment.

Thirty age-matched normal-weight Italian individuals (5 men and 25 women; mean age ± SD: 32.1 ± 6.86 years; mean Body Mass Index (BMI: 21.8 ± 2.90 kg/m^2^) served as controls. The mean age between the two samples was not significant (t = 1.89; *p* = 0.06), while the mean BMI was significantly higher in PWS than in normal-weight individuals (*p* < 0.001).

Participants with PWS were consecutively recruited at the Division of Auxology, IRCCS Istituto Auxologico Italiano, an Italian third-level clinical center for subjects with PWS. Only participants of both sexes, older than 18 years of age, with a genetically confirmed diagnosis of PWS who achieved a score >24 in the Mini-Mental State Examination (MMSE) [11], which warranted an intellectual level allowing appropriate compliance, were included. Physical or intellectual disabilities that could compromise participation in the study or any psychiatric disorders were exclusion criteria. For the control group, participants were recruited using an in-person method among the hospital, medical, research, and administrative staff, as well as friends and colleagues. Convenience sampling was utilized due to its practicality in accessing readily available populations; however, this approach inherently limits generalizability. Efforts were made to mitigate these limitations by targeting subjects of the same age, sex, and social status. Participants were eligible if they were males/females older than 18 years with normal weight (BMI from 18.5 to 24.9) and without any form of physical or intellectual disability that could compromise their participation in the study. The control group was selected after a careful anamnestic collection, excluding those subjects with psychiatric and medical problems.

### 2.2. Procedures

Once informed about the study and after obtaining written informed consent to participate from all subjects and their parents or legal guardians, participants were screened for eligibility criteria at their admission to the Institute. Then, all the participants were asked to fill out a self-report questionnaire described below to assess our variables of interest. The administration of MSSE was made under the supervision of a member of our research team, which verified the patient’s understanding of the questions. Data were collected from May 2023 to March 2024. This study was approved by the Ethical Committee of Istituto Auxologico Italiano, IRCCS, Milan, Italy (approval number: CE: 2023_03_21_02; research code: 01C310; acronym: PROPSICOPWS). The research was carried out according to the Declaration of Helsinki and its advancements.

### 2.3. Measures

For patients with PWS, demographical variables were obtained from parents (and/or main caregivers) and patients. The internal medical staff measured weight and height. Standing height was determined by a Harpenden Stadiometer (Holtain Limited, Crymych, Dyfed, UK). Weight was measured to the nearest 0.1 kg using an electronic scale (RoWU 150, Wunder Sa.bi., Trezzo sull’Adda, Italy). Body Mass Index (BMI) was calculated using the formula kg/m^2^.

The Mini-Mental State Evaluation (MMSE; 11) was performed to assess cognitive functions. It assesses the following cognitive domains: Orientation, awareness of time, place, and personal details (e.g., “What is the date today?”); Registration, the ability to repeat a set of words immediately after hearing them; Attention and Calculation: tasks such as spelling a word backward or performing serial subtractions; Recall: recalling previously stated words after a brief delay; Language and Praxis: naming familiar objects, following written or spoken commands, writing a sentence, copying a design.

Clinical data about the quality of life and psychological well-being were obtained via the administration of the Italian versions of the 36-item Health Survey Short Form-36 (SF-36) [12,13] and the Psychological General Well-Being Index (PGWBI) [14,15].

SF-36 is a self-report questionnaire composed of 36 items assessing the quality of life in several domains: Physical functioning: limitations in physical activities, such as walking or climbing stairs (e.g., Does your health now limit you in these activities? If so, how much?); role physical: the impact of physical health on work and daily role performance (e.g., “During the past 4 weeks, how much of the time have you had problems with your work or other regular daily activities as a result of your physical health?”); bodily pain: pain intensity and its interference with normal activities (e.g., “How much bodily pain have you had during the past 4 weeks?”); general health: perceptions of overall health and expectations of future health (e.g., “I seem to get sick a little easier than other people.”); vitality: energy levels and fatigue (e.g., “How much of the time during the past 4 weeks did you feel worn out?”); social functioning: the impact of physical or emotional health on social activities (e.g., “During the past 4 weeks, how much of the time have your physical health or emotional problems interfered with your social activities ?”); role emotional: limitations in daily roles due to emotional problems (e.g., “During the past 4 weeks, how much of the time have you had problems with work or other regular daily activities as a result of any emotional problems (such as feeling depressed or anxious)?”); well-being: general emotional well-being, including symptoms of anxiety and depression (“How much of the time during the past 4 weeks have you felt calm and peaceful?”). Scores for each domain range from 0 to 100, with higher scores indicating better health or functioning.

The PGWBI is a self-report questionnaire composed of 22 items rated on a six-point Likert scale ranging from 0 to 5, assessing six dimensions of psychological well-being: Anxiety: feelings of nervousness, worry, and unease (e.g., “I felt anxious or nervous”); depression: feelings of sadness, hopelessness, and lack of enjoyment (e.g., I felt that life was meaningless”); positive well-being: feelings of happiness, life satisfaction, and optimism (e.g., “I felt cheerful, lighthearted”); self-control: perceived control over emotions and behavior (e.g., “I was emotionally stable and sure of myself”); general health: perceptions of physical health and resilience (e.g., “I felt physically fit and well”); vitality: energy levels and feelings of fatigue or lethargy (e.g., ”I felt full of energy”). Scores can be calculated for each domain or summed for a total score, providing an overall measure of psychological well-being. The total score ranges from 0 to 110. Higher scores indicate greater well-being.

### 2.4. Statistical Analysis

Descriptive statistics of the sample were conducted to assess frequencies and percentages for categorical variables and means and standard deviations for continuous variables. The normal distribution of the variables was assessed by skewness and kurtosis indices. The normal distribution was guaranteed if skewness and kurtosis indices were within the acceptable range between −2 and +2 [16]. To compare data on SF-36 and PGWBI of adult individuals with PWS with normal-weight subjects, independent sample t-tests were performed comparing the scores on each subscale of SF-36 and PGWBI. Cohen’s *d* effect size was also calculated. Correlations between similar subscales of both questionnaires involved (vitality, positive well-being, and general health of PGWBI and vitality, well-being, and general health of SF-36) were also explored using Pearson’s correlation coefficient. Critical alpha was set at 0.05.

## 3. Results

Means, standard deviations, and t-tests for all subscales of SF-36 and PGWBI are depicted in Table 1.

The comparisons between our sample of Italian adult patients with PWS and the sample of age-matched Italian individuals on SF-36 showed that the physical health (*p* < 0.001) and the social functioning (*p* = 0.04) subscales were significantly higher in normal-weight individuals than in patients with PWS. On the contrary, the vitality subscale was higher in patients with PWS than in the control group (*p* < 0.001). No other significant differences were found in SF-36 between patients with PWS and normal-weight individuals.

As far as PGWBI is concerned, data showed that the vitality subscale of PGWBI was significantly higher in patients with PWS than in normal-weight individuals (*p* = 0.01), while the Self-Control subscale of PGWBI was significantly higher in normal-weight individuals than in patients with PWS (*p* = 0.05). No other significant differences were found in PGWBI between patients with PWS and normal-weight individuals.

Correlations between similar subscales of the two questionnaires involved in the study revealed that the vitality subscale of SF-36 was positively and significantly correlated with the vitality subscale of PGWBI, the well-being subscale of SF-36 was positively and significantly correlated with the positive well-being subscale of PGWBI, and the general health subscale of the SF-36 was positively and significantly correlated with the general health subscale of the PGWBI subscale. Correlations were also performed for each group. In the group of patients with PWS, the vitality subscale of PGWBI and the vitality subscale of SF-36 were positively and significantly correlated, while the positive well-being subscale of PGWBI was not significantly correlated with the well-being subscale of SF-36, and the general health subscale of PGWBI was positively and significantly correlated with the general health subscale of the SF-36. In the group of normal-weight individuals, the vitality subscale of PGWBI and the vitality subscale of SF-36 were positively and significantly correlated, the positive well-being subscale of PGWBI was positively and significantly correlated with the well-being subscale of SF-36, and the general health subscale of PGWBI was positively and significantly correlated with the general health subscale of the SF-36.

Correlations are presented in Table 2.

Although the relatively small sample size of PWS patients in this study did not allow us to subdivide participants into BMI-specific subgroups, we also conducted correlational analyses between BMI and the total scores of the PGWBI and SF-36 scales. No significant correlations were found between BMI and the total PGWBI score. The only correlations identified were between BMI and the general health (*p* = 0.01) and vitality (*p* = 0.04) subscales of the SF-36. 

## 4. Discussion

This study aimed to assess the quality of life and psychological well-being in a sample of Italian adult patients with PWS compared to age-matched normal-weight Italian individuals. As expected, the results of the study indicated that quality of life and psychological well-being in patients with PWS were impaired in several aspects. These included the perception of limitations in physical activities, restrictions in social functioning due to their physical and emotional problems, and reduced perceived control over emotions and behaviors. Specifically, scores on the physical health and social functioning subscales of the SF-36 and the Self-Control subscale of the PGWBI were significantly lower in PWS patients compared to their normal-weight counterparts. In those scales, the effect sizes were medium to large, thus suggesting that the differences between groups were significant. These findings align with previous evidence suggesting substantial impairment in both physical and mental health among patients affected by PWS [7,17]. Caregivers of PWS patients have also reported significant challenges in their children’s quality of life, particularly in social and school functioning and physical health domains [9].

Conversely, this study found that patients with PWS showed higher vitality levels than their age-matched normal-weight counterparts, suggesting a preserved energy level during the day. The effect size for this difference was medium, indicating a moderate disparity between groups.

Similar patterns were also observed in a previous study by our research group comparing SF-36 and PGWBI scores of PWS patients to normative data from the Italian population [13,15]. In that study, PWS patients scored significantly lower on the physical health and social functioning subscales of the SF-36 compared to normative subjects but higher on the general health, vitality, and well-being subscales. Regarding the PGWBI, PWS patients demonstrated significantly higher scores on the total score, positive well-being, general health, and vitality subscales than normative data, reflecting better perceived well-being. However, their scores on the Self-Control subscale were lower than those of the normative sample.

A plausible explanation for the unexpected and apparently contradictory findings about the psychological well-being of patients with PWS could be reasonably explained by their living conditions. PWS patients often live with their families in highly controlled environments, receive regular medical care, and benefit from substantial familial, medical, and social support. These factors may create a protective framework that prevents them from experiencing significant negative events, thus contributing to a sense of well-being. Supporting this hypothesis, a narrative medicine study exploring perceptions of daily life among PWS patients and their caregivers revealed that patients often described themselves as feeling good, mostly happy, and proud, with occasional feelings of sadness about their condition [18]. Additionally, a recent study by Cobo and colleagues [19] noted that while patients with PWS had good awareness of their illness and of the effects of treatment, they also lacked a clear understanding of the long-term consequences of the disease.

This limited awareness could partly explain why they report feeling good in their daily lives despite their compromised health conditions.

However, further research is needed to explore these findings more comprehensively. The discrepancies in the literature regarding the psychological well-being of PWS patients could stem from differences in study design, sample characteristics, and assessment methods. To address these inconsistencies, longitudinal studies tracking psychological health over time, comparative studies examining psychological profiles across genetic subtypes, cultural, and social contexts, or meta-analyses aggregating data from multiple studies would provide valuable insights. These approaches could help identify overarching trends and factors contributing to the observed differences.

### 4.1. Strengths and Limitations

The strengths and limitations of this study warrant discussion. One notable strength is that it represents one of the few studies focusing on the quality of life and psychological well-being of adult patients with PWS, providing a valuable comparison between a clinical sample of Italian adults with PWS and age-matched Italian individuals from the general population.

Regarding the comparison between patients with PWS and age-matched normal-weight Italian individuals, it should be acknowledged that the two groups differed slightly in terms of gender composition and average age. However, previous studies have demonstrated that gender and age do not significantly influence responses to the SF-36 and PGWBI, making these differences negligible [20,21]. This study was then conducted in a third-level specialized clinical center, which allowed us to recruit a relatively numerous study group of subjects with this rare disease. This study also used widely adopted and well-validated self-report questionnaires.

Nonetheless, this study has limitations. Its cross-sectional design precludes conclusions about causality between the variables examined. Furthermore, the use of self-report measures introduces potential biases, such as socially desirable responses. In addition, the number of analyses may improve the risk for Type 1 error, thus incorrectly rejecting the null hypothesis when it is true. Importantly, it remains unclear whether the psychological states described in this study are attributable to the syndrome itself or are influenced by Body Mass Index (BMI). Addressing this question would require comparing adequately sized subgroups of PWS patients with normal weight, overweight, and obesity. Another informative comparison could involve age- and BMI-matched individuals with essential overweight or obesity.

Although the relatively small sample size of PWS patients in this study did not allow us to subdivide participants into BMI-specific subgroups, we conducted correlational analyses between BMI and the total scores of the PGWBI and SF-36 scales. No significant correlations were found between BMI and the total PGWBI score. The only correlations identified were between BMI and the general health (*p* = 0.01) and vitality (*p* = 0.04) subscales of the SF-36. These preliminary findings, while intriguing, require confirmation through future studies involving larger populations.

### 4.2. Clinical Implications

Despite the limitations mentioned above, this study contributes to expanding the knowledge of the psychological conditions of patients with PWSA. A deeper understanding of the specific psychological challenges faced by individuals with PWS will enable clinicians to design targeted therapeutic interventions tailored to their unique needs.

### 4.3. Conclusions

These findings underscore the importance of continued research into the psychological dimensions of PWS and the implementation of policies that ensure equitable access to comprehensive mental health services for this population. Future studies should investigate how various multidisciplinary interventions might influence long-term psychological and functional outcomes. Such research will help optimize treatment strategies and improve the quality of life for individuals living with PWS.

## Figures and Tables

**Table 1 healthcare-13-00158-t001:** Means, standard deviations, and t-tests for all subscales of SF-36 and PGWBI.

	PWS Sample (N = 30)		Normal-Weight Sample (N = 30)		t	*p*	Cohen’s *d*
	M	SD	M	SD			
SF-36 Physical Health	65.0	34.42	97.8	4.49	−5.18	<0.001 *	−1.34
SF-36 Role Physical Health	76.7	32.12	89.2	24.29	−1.70	0.095	−0.44
SF-36 Bodily Pain	81.4	31.46	88.7	20.99	−1.05	0.299	−0.27
Sf-36 General Health	74.5	21.23	73.2	16.84	0.27	0.788	0.07
SF-36 Vitality	79.3	22.08	60.3	16.97	3.74	<0.001 *	0.96
SF-36 Social Functioning	63.8	38.47	80.0	20.39	−2.04	0.047 *	−0.53
SF-36 Emotional Problems	80.0	28.50	81.1	31.18	−0.14	0.886	−0.04
SF-36 Well-Being	78.4	25.25	75.7	17.00	0.48	0.633	0.12
PGWBI Anxiety	19.1	6.74	17.8	4.42	0.91	0.369	0.24
PGWBI Depression	12.4	4.01	13.4	1.59	−1.35	0.181	−0.35
PGWBI Positive Well-being	15.0	4.44	14.0	3.20	1.07	0.291	0.27
PGWBI Self-Control	10.4	3.75	12.0	2.67	−1.95	0.057 *	−0.50
PGWBI General Health	12.5	3.14	12.4	1.75	0.20	0.840	0.05
PGWBI Vitality	15.9	4.49	13.2	3.12	2.67	0.010 *	0.69
PGWBI Total Score	85.2	21.85	82.7	14.00	0.53	0.600	0.14

* significant differences.

**Table 2 healthcare-13-00158-t002:** Correlations between similar subscales of the two questionnaires for each group.

PWS Group	Pearson’s r Coefficient	*p*
SF-36 Vitality—PGWBI Vitality	0.44	0.014 *
SF-36 Well-Being—PGWBI Positive Well-Being	0.67	0.091
SF-36 General Health—PGWBI General Health	0.31	<0.001 *
Normal-weight group		
SF-36 Vitality—PGWBI Vitality	0.82	<0.001 *
SF-36 Well-Being—PGWBI Positive Well-Being	0.88	<0.001 *
SF-36 General Health—PGWBI General Health	0.48	<0.001 *

* significant correlations.

## Data Availability

Raw data will be uploaded on www.zenodo.org immediately after the manuscript is accepted, and they will be available upon reasonable request from the authors A.G.U. and A.S.

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
