# Peer review of "Assessment of Quality of Life and Psychological Well-Being in Italian Adult Subjects with Prader–Willi Syndrome Using the Health Survey Short Form and the Psychological General Well-Being Index Questionnaires"

_healthcare, 2025, doi:10.3390/healthcare13020158_

Round 1
Reviewer 1 Report
Comments and Suggestions for Authors
The article investigates the perception of well-being of a group of Italian patients with Prader-Willi syndrome (PWS) compared with a control group. Although the research concerns a specific group of carriers of a rare genetic syndrome, it is interesting because it attempts to contribute to the knowledge of the perception of well-being in PWS patients. The choice of subjects that make up the two groups compared is well documented and reflects the standards that scientific research must have. The lack of gender balance, indicated by the authors as a possible limitation of the study, is amply justified in the literature and does not represent a flaw. From a methodological point of view, an accurate description of the instruments used to measure subjective well-being is missing, e.g. the composition of the subscales and their range. It would be appropriate to spend a few more lines also because the results refer to the correlations between subscales of different instruments. To understand the meaning of the values ​​found it is better to have a more in-depth knowledge of the construction criteria of the subscales. The measurements collected are not particularly numerous but they were analyzed correctly by verifying the adequacy of the distributions and taking into account the small number of the two groups. The results, the discussion of which could be more detailed, allow us to integrate existing knowledge on patients suffering from PWS with particular reference to the Italian context.
Comments on the Quality of English LanguageAlthough formally correct, it seems to refer to a non-English construct: a review by a native speaker could further improve its quality.
Reviewer 2 Report
Comments and Suggestions for Authors
Thank you for the paper. Please find comments how to improve it.
1. Please provide more specific conclusions in the abstract.
2. Please add more keywords (up to 10) for better indexing.
3. Please unify the style of writing the numbers: "thirty patients with PWS and 30".
4. Please add paragraphs in 2.1. Participants and Procedures, making text more structured.
5. Please indicate when the study was conducted.
6. Please present strategies to limit the bias in the control group recruitment.
7. Please describe questionnaires used in more detail. Please add examples of items in each subscale. The MMSE should also be described.
8. Effect size should be calculated. Please discuss the effect size in the discussion section.
9. Table 1 and 2 could be merged. There is no need to present these data in two tables.
10. All correlation should be presented in a form of a table. Also, this could be done separately for clinical and non-clinical groups.
11. Please restructure the discussion by adding specific section like Limitations and Practical implications.
12. Conclusions should be presented in a separate section.
Overall, the paper seems to very simple methodology with simple analysis and hypotheses, and this might be not enough for the Healthcare journal. However, it has been prepared accurately. Please enrich your paper by the suggestions presented above. Please also present more ideas how these results can be incorporated in healthcare policies.
Reviewer 3 Report
Comments and Suggestions for Authors
The study addresses the quality of life and well-being in adults affected by Prader-Willi syndrome, with the aim of filling the gaps in the Italian literature on these topics. But the authors seem not to really hit their target.
In fact, the study presents some organizational/descriptive/methodological gaps:
i. Authors affirm: “As expected, the results of the study showed that quality of life and psychological well-being in patients with PWS were impaired in many aspects including physical, behavioral, and social aspects”. But only two of eight SF-36 scales were significantly lower for PWS people when compared with the same scores of normal-weight individuals. Furthermore, in three scales (i.e., Vitality with significative differences, General health, and Well-being) the scores of PWS people were upper than the same scores of normal-weight participants: so, the quality of life of normal weight-participants were not well too, when globally considered!
ii. The real insurmountable gap is the lack of data on the health and care conditions of people with PWS. For example, in order to assess psychiatric impairments cited by Authors, only anxiety and depression indexes were detected through PGWBI and no information is given about care (e.g., hormonal, etc.) or referred to psychological support or other. In summary, it’s impossible to predict the variances of the SF-36 and PGWBI scale scores and so we could not know what factors explained the measures of the quality of life and well-being. The MBI appears to be the only relevant factor.
iii. Finally, Authors state that their results "provide valuable insights useful to improve the psychological approaches for the multidisciplinary treatment of PWS", but in the Discussion section this topic is not developed.
I consider the study not publishable: I found interesting that SF-36 Vitality was higher in people with PWS than in normal-weight people, but with the methodology used it cannot explain why this occurred.
Round 2
Reviewer 2 Report
Comments and Suggestions for Authors
Great job!
Just presenting a minor suggestion: Please round the numbers up using two decimal places except p-values, which should be presented with three decimal places (see for instance, Tables 1 and 2).
Once again, great improvements!
Reviewer 3 Report
Comments and Suggestions for Authors
This revised version of the article is much better than the previous one and is publishable. It helps fill an important gap in the literature referring to Italian PWS patients
However, some revisions are still needed.
First, no information is given on any psychological treatments for PWS patients, which could explain the unexpected contradictory results.
Secondly, other minor revisions are detailed below.
Finally, I do not see that "These findings ... provide valuable insights for improving multidisciplinary psychological treatment approaches for these individuals", so I would either omit this and the other statements in the Conclusion section or explain in more detail how this could happen.
Minor revisions: lines 269 & 427 – I think that “MSSE” should be MMSE
RESULTS: lines 414-433 – I think that the description of the sample characteristics has to be placed in “Participants” section.
Table 1: please, write ‘t’ instead of ‘T’; “*” note is redundant, the p value is sufficient; the p value = .057 is not significant (it’s = .06!).
As a test of significance, two MANOVAs (one for SF-36 scales and one for PGWBI scales) are preferable instead of 15 t-tests, one for each scale, because the subscales of each test are not completely independent from each other.
Line 481: I don't understand the sentence "The results were the same regardless of family composition and the presence of comorbid conditions in patients with PWS.": how else could the results have been different? – Please, clarify or omit.
Lines 495-512 and Table 2 – The correlations coefficients reported in the test and in the table together are redundant (so * note in the table) – please, simplify.
DISCUSSION: lines 700 and beyond: please, report BMI-scales correlations in result section; why did not?
